# Caspofungin Cerebral Penetration and Therapeutic Efficacy in Experimental Cerebral Aspergillosis

Irina Ullmann,[a,c] Andrea Aregger,[a,d] Stephen L. Leib,[a] Stefan Zimmerli[a,b]

[a]Institute for Infectious Diseases, University of Bern, Bern, Switzerland
[b]Department of Infectious Diseases, Bern University Hospital, University of Bern, Bern, Switzerland
[c]Clinic of General Internal and Emergency Medicine, Citizens Hospital Solothurn, Solothurn, Switzerland
[d]Center for Intensive Care Medicine, Lucerne Cantonal Hospital, Lucerne, Switzerland

Irina Ullmann and Andrea Aregger contributed equally to this article. Author order was determined in order of increasing seniority.

**ABSTRACT** Despite best available therapy, cerebral aspergillosis is an often-lethal complication of disseminated aspergillosis. There is an urgent need to expand the currently limited therapeutic options. In this study, we assessed cerebral drug exposure and efficacy of caspofungin (CAS) using a lethal infant rat model of cerebral aspergillosis. Eleven-day-old Wistar rats were infected by intracisternal injection of *Aspergillus fumigatus* conidia. Treatment started after 22 h and was continued for 10 days. Regimens were CAS 1 mg/kg/day intraperitoneally (i.p.), liposomal amphotericin B (L-AmB) 5 mg/kg/day i.p., and both drugs combined at the same dose i.p. Infected controls were given NaCl 0.85% i.p. Primary endpoints assessed were survival, cerebral fungal burden, galactomannan index, and drug concentrations in brain homogenate at 2, 3, 5, and 11 days after infection. Compared to those of controls ($4.4 \pm 2.7$ days), survival times were increased by treatment with CAS alone ($10.3 \pm 1.7$ days; $P < 0.0001$) and CAS combined with L-AmB ($9.3 \pm 2.8$ days; $P < 0.0001$). In contrast, survival time of L-AmB-treated animals ($4.3 \pm 3.1$ days) was not different from that of controls. Cerebral fungal burden and galactomannan index declined in all animals over time, without significant differences between controls and treated animals. CAS trough levels in brain tissue were between 0.84 and 1.4 $\mu$g/g, concentrations we show to be associated with efficacy. AmB trough levels in brain tissue were higher than the MIC of the *A. fumigatus* isolate. In summary, CAS concentrations in brain tissue suggest it may be therapeutically relevant and it significantly improved survival in this lethal model of cerebral aspergillosis in nonneutropenic rats. The clinical efficacy of CAS treatment for cerebral aspergillosis merits further study.

**IMPORTANCE** Treatment options for cerebral aspergillosis, an often-lethal disease, are limited. The echinocandins (caspofungin is one of them) are not recommended treatment because their brain tissue penetration is often considered insufficient. In a nursing rat model of cerebral aspergillosis that mimics human disease, we found potentially therapeutically relevant concentrations of caspofungin in brain tissue and prolonged survival of caspofungin-treated animals. The efficacy of caspofungin in the treatment of cerebral aspergillosis documented here, if confirmed in other animal models (especially immunosuppressed murine models) and by using additional *Aspergillus* isolates across a range of CAS minimal effective concentrations (MECs), would suggest that caspofungin merits further study as a treatment option for patients suffering from aspergillosis disseminated to the brain.

**KEYWORDS** cerebral aspergillosis, antifungal therapy, caspofungin, CNS drug penetration, animal model, rat

Address correspondence to Stefan Zimmerli, stefan.zimmerli@insel.ch.

The authors declare no conflict of interest.

10.1128/spectrum.02753-21 **1**

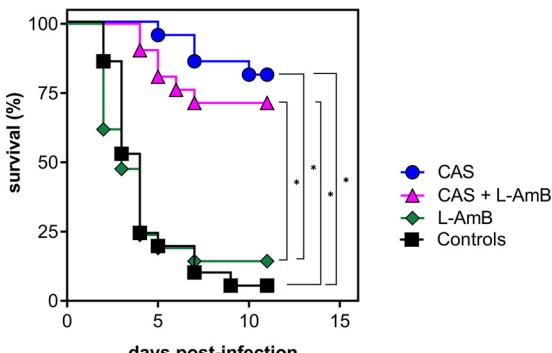

**FIG 1** Comparison of survival curves for nonimmunosuppressed infant rats infected with 7.18 $\log_{10}$ CFU *A. fumigatus* conidia by intracisternal injection. Treatment was started at 22 h postinfection and continued for 10 days with daily doses of CAS 1 mg/kg i.p., L-AmB 5 mg/kg i.p., and CAS plus L-AmB combined at the same dose i.p. Controls were given NaCl 0.85% i.p. Results of 3 independent experiments are shown ($n = 21$ per treatment group). *, $P < 0.05$ compared to controls and to L-AmB. In contrast to all other treatment regimens, L-AmB treatment did not prolong survival compared to controls.

Invasive aspergillosis is a leading cause of illness and death among severely immunocompromised patients (1, 2). It primarily affects the lungs, the usual portal of entry of aerosolized *Aspergillus* conidia. Cerebral aspergillosis, which is observed in 14 to 42% of cases, is the most common organ manifestation of hematogenous dissemination from pulmonary foci (3–6). Before the introduction of mold-active triazole drugs with enhanced penetration in brain tissue, cerebral aspergillosis was considered to be nearly universally fatal, despite treatment (3). With voriconazole therapy, 1-year survival rates of 31% have been reported (7). Limited experience is available on brain tissue exposure and therapeutic efficacy for cerebral aspergillosis of isavuconazole, a mold-active triazole drug recently approved for the treatment of invasive aspergillosis (8). The echinocandins, the third class of mold-active antifungal drugs, are not recommended for the treatment of central nervous system (CNS) aspergillosis (9). Despite their efficacy in murine and rat models of cerebral aspergillosis, their brain tissue penetration is often considered insufficient, although data on brain tissue exposure are scarce and experience with echinocandin therapy of human CNS aspergillosis is limited (10–13).

Here, we report on the cerebral penetration and treatment effect of the echinocandin caspofungin (CAS) in a model of cerebral aspergillosis in nonimmunosuppressed infant rats (14).

## RESULTS

**Effect of antifungal treatment on survival rates.** The survival rate of untreated control animals and animals treated with antifungal drugs alone or in combination was monitored for the first 11 days after infection. Figure 1 presents the Kaplan-Meier plots of cumulative mortality for each of the treatment regimens. The model proved to be highly lethal, with 20 of 21 control animals succumbing within 11 days after infection. Mean survival time of untreated controls ($n = 21$) was 4.3 ± 2.4 days. Treatment with CAS alone ($n = 21$) and combined with L-AmB ($n = 21$) significantly increased survival time to 10.3 ± 1.7 days ($P < 0.0001$) and 9.3 ± 2.8 days ($P < 0.0001$), respectively. There was no significant difference between these 2 treatment regimens. In contrast, the survival curve of animals treated with L-AmB alone ($n = 21$) declined rapidly with a mean survival time of 4.3 ± 3.1 days, not significantly different from that of controls ($P = 0.47$). Interestingly, the lowest mortality rate with 17 of 21 animals surviving until day 11 was observed with treatment with CAS, a drug that is not considered effective for treatment of cerebral aspergillosis in patients.

**Effect of antifungal treatment on cerebral fungal load.** To measure cerebral fungal burden, quantitative fungal cultures were done. The recovery of *Aspergillus*

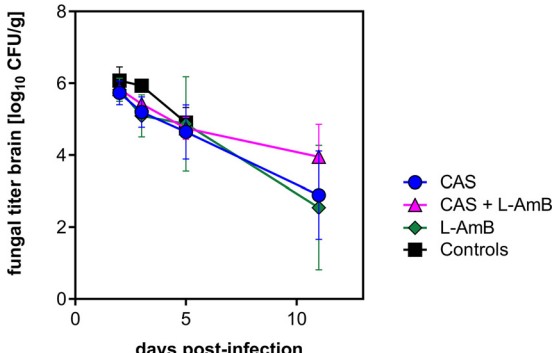

**FIG 2** Comparison of the cerebral fungal titer (indicated as $\log_{10}$ CFU/g of brain tissue) of nonimmunosuppressed infant rats infected with 7.18 $\log_{10}$ CFU *A. fumigatus* conidia by intracisternal injection. Results of 3 independent experiments ($n = 21$ for active treatment; $n = 11$ for controls) are shown. Values are mean ± standard deviation (SD). The cerebral fungal burden measured by quantitative fungal culture declined over time in all animals, including untreated ones. There was no significant difference between controls and treated animals.

*fumigatus* from the brains is graphed in Fig. 2. Following intracisternal infection with 7.18 $\log_{10}$ CFU *A. fumigatus* conidia, all brains yielded positive cultures. Maximal growth of 6.08 ± 0.38 $\log_{10}$ CFU/g was found in control animals 2 days after infection. The numbers of *Aspergillus* CFU in brain homogenates declined over time in all animals, including untreated ones, and reached a minimum of 2.54 ± 1.74 $\log_{10}$ CFU/g in L-AmB-treated animals 11 days after infection. Interestingly, there was no significant difference between controls and treated animals ($P = 0.18$).

**Effect of antifungal treatment on galactomannan index.** Because CFU counts may not be an accurate measure of hyphal growth in tissue and of the antifungal effect of CAS that affects only the growing part of mycelia, we also determined the galactomannan index (GMI) in homogenized brain tissue as a measure of hyphal mass. The GMI over time is depicted in Fig. 3. Except in L-AmB-treated animals, which reached maximal values on day 2, GMI peaked on day 3, a day later than CFU counts, in all other treatment groups, including controls, and declined over the further course of the experiment, similar to the CFU count. As was observed for CFU counts, there was no significant difference between controls and treated animals ($P = 0.09$).

**Drug levels.** To investigate whether the administered antifungal drugs reached potentially therapeutic concentrations in brain tissue, drug levels were determined by high-pressure liquid chromatography (HPLC). Cerebral drug levels for CAS and AmB are shown in Fig. 4. Since central nervous system aspergillosis is associated with a disrupted brain blood barrier, we found CAS trough concentrations in brain tissue between 0.84 ± 0.24 $\mu$g/g on day 2 and 1.4 ± 0.54 $\mu$g/g on day 5 of the experiment.

The AmB trough concentrations in brain tissue ranged among 0.68 ± 0.42 $\mu$g/g on day 2, 1.05 ± 0.62 $\mu$g/g on day 3, and 0.87 ± 0.36 $\mu$g/g on day 11. There were no statistically significant differences over time.

## DISCUSSION

Central nervous system aspergillosis is a highly fatal disease, despite best available therapy. It is the most common organ manifestation of disseminated invasive infection, usually via the bloodstream from a pulmonary focus, occurring in up to 42% of patients immunocompromised because of hematological malignancies (4, 6, 15). Direct invasion of the brain from adjacent foci of aspergillosis in paranasal sinuses or the middle ear is a rare alternative route of infection that may occur in immunocompetent hosts. Voriconazole combined with neurosurgical debridement, where feasible, is the recommended first line of treatment for CNS aspergillosis (9, 16). The high failure rate of this approach, the potential for drug-drug-interactions, and the worldwide emergence of azole-resistant *Aspergillus* isolates make new therapeutic approaches highly desirable (17, 18). CAS is currently not recommended as primary treatment for invasive aspergillosis, and several guidelines recommend

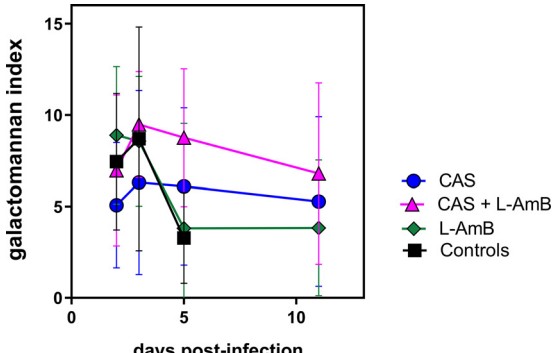

**FIG 3** Comparison of the galactomannan index of nonimmunosuppressed infant rats infected with 7.18 log$_{10}$ CFU *A. fumigatus* conidia by intracisternal injection. Results of 3 independent experiments (*n* = 21 for active treatment; *n* = 11 for controls) are shown. Values are mean ± SD. The galactomannan index peaked on day 2 in L-AmB-treated animals and on day 3 in all other animals and then declined over time in all animals, including untreated ones. There was no significant difference between controls and treated animals.

against its use for CNS aspergillosis because of uncertainty about tissue penetration (9, 16). Clinical data on CAS brain tissue penetration, however, are not available. In animal models of CNS aspergillosis, CAS treatment significantly reduced mortality and intracerebral fungal burden. Cerebral deposition of CAS was not measured in these studies (10, 11, 13). The aim of this study was to determine the therapeutic efficacy and cerebral penetration of CAS using a lethal model of cerebral aspergillosis in nonimmunosuppressed nursing rats (14).

We found that CAS is effective in reducing mortality and that it reaches high concentrations in brain tissue. Compared to that of untreated controls, survival was significantly prolonged with CAS infection, from 4.3 to 10.3 days, and CAS protected 17 out of 21 animals from mortality. We did not determine whether treatment with CAS was curative in this model. Treatment with L-AmB alone, in contrast, had no effect on survival. Combining L-AmB treatment with CAS resulted in survival rates comparable to those of monotherapy with CAS.

The efficacy of CAS in prolonging survival in nursing rats is in accordance with findings in a model of cerebral aspergillosis in cyclophosphamide-treated mice where CAS administered for 10 days resulted in significant reduction of mortality, similar to L-AmB (10, 11). In the same model, CAS was equally as effective as L-AmB in reducing the cerebral fungal burden (13). In a model of systemic aspergillosis in cyclophosphamide-immunosuppressed mice, CAS was effective in prolonging survival. Similar to our findings, compared to that of the control, survival was not increased by L-AmB, and

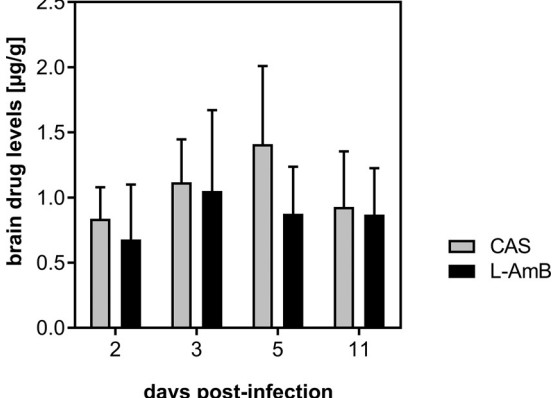

**FIG 4** Drug levels of caspofungin (CAS) and AmB (L-AmB) determined by HPLC in brain homogenate at various time points after infection. Results of 2 independent experiments are shown (*n* = 21 per treatment group). Over time, no statistically significant differences in drug levels were observed for CAS and AmB.

addition of CAS to L-AmB salvaged animals from the high mortality observed with L-AmB alone (19).

In many experimental models of invasive aspergillosis, irrespective of the method of immunosuppression, prolonged survival was associated with a reduction in fungal tissue burden (20–22). In the present model, in contrast, there was no correlation between survival rate and cerebral fungal burden measured by CFU count and by determination of the GMI. This finding may, in part, be explained by the absence of immunosuppression and the maturation of the immune response over the course of the experiment. In untreated animals, there is a vigorous inflammatory reaction around dense cerebral foci of fungal growth (14). Differential immunomodulatory effects of CAS and L-AmB may, in addition to direct antifungal action, affect the survival difference between animals with similar fungal burdens (19, 23, 24). Indeed, in a murine model of invasive aspergillosis that compared different immunosuppressive regimens, AmB did not improve survival over that of the control in corticosteroid-treated animals where disease pathology is driven by inflammation. In contrast, AmB was effective treatment for neutropenic animals where the disease process was driven by uncontrolled fungal growth (22). Immunomodulatory effects of CAS in experimental aspergillosis have been shown to enhance human neutrophil-associated damage to the molds (25, 26). In contrast, L-AmB immunomodulation in systemic aspergillosis includes upregulation of proinflammatory cytokines, which was associated with poor survival in a murine model (19).

Because of its large molecular weight (1,093.3 Da) and its high level of protein binding, CAS deposition in brain tissue is considered to be low (27). In uninfected rats, CAS brain tissue concentrations following an intravenous bolus of 2 mg/kg were below 0.17 $\mu$g/g and showed little variation over the first 24 h (28). A CAS brain tissue-to-plasma ratio of approximately 20% was found in a model of immunocompetent, noninfected adult rats (29). We measured CAS trough levels in brain tissue between 0.84 and 1.4 $\mu$g/g, concentrations we show to be associated with efficacy.

Disruption of the blood-brain barrier due to cerebral abscesses is observed in patients and reflected in our model and may have contributed to the high tissue CAS concentrations (14). Similarly, in a murine model of CNS candidiasis, CAS brain levels were higher in infected animals than in noninfected animals. In this murine model, after 7 days of daily dosing, concentration of CAS in brain tissue was relatively constant over 24 h, indicating that half-life was longer in brain tissue than in plasma (30).

We found AmB trough concentrations in brain tissue between 0.78 and 1.05 $\mu$g/g without significant variation over the course of the experiment. Considering that for AmB concentration-dependent indices are correlated closely with efficacy, insufficient drug levels do not appear to explain the lack of efficacy of L-AmB in our model (31). Indeed, the trough levels of AmB achieved in brain tissue were comparable to those in a rat model of systemic aspergillosis where repeat daily dosing of L-AmB at 5 mg/kg resulted in brain trough levels of 0.7 $\mu$g/g and successfully reduced fungal titers in all organs, including the brain (32).

In summary, in a nursing rat model of cerebral aspergillosis, we found potentially therapeutically relevant concentrations of CAS in brain tissue and prolonged survival of CAS treated animals. The efficacy of CAS in the treatment of cerebral aspergillosis documented here, if confirmed in other animal models (especially immunosuppressed murine models) and by using additional *Aspergillus* isolates across a range of CAS minimal effective concentration (MECs), would suggest that CAS merits further study as a treatment option for patients suffering from aspergillosis disseminated to the brain.

## MATERIALS AND METHODS

**Fungal inoculum.** The strain of *Aspergillus fumigatus* used was isolated from a patient with invasive pulmonary aspergillosis (14). Broth dilution susceptibility testing (CLSI M38-A) indicated that it was susceptible to all antifungal drugs used in the experiments (33). The MICs were amphotericin B (AmB) 0.5 mg/L, itraconazole 0.016 mg/L, voriconazole 0.25 mg/L, and posaconazole 0.032 mg/L. The MEC for CAS was 0.016 mg/L. The isolate stored in distilled water at 4°C was grown on Sabouraud's agar plates with chloramphenicol and gentamicin for 3 days at 37°C prior to collection of conidia. Conidia were

harvested by washing the agar surface with sterile phosphate-buffered saline (PBS) containing 0.05% Tween 80 and filtered through a BD Falcon 40 $\mu$m nylon cell strainer (BD Biosciences, Bedford, MA, USA) to remove hyphal fragments and conidial clumps. The resulting stock suspension of conidia was adjusted using a hemacytometer, and the number of conidia was determined by plating on Sabouraud's agar plates with chloramphenicol and gentamicin and counting the CFU after 24 h at 37°C. Three days before infection, the final concentration was adjusted and verified by CFU counts. In addition, the fungal titer of the inoculum was routinely confirmed by quantitative culture on the day of infection.

**Infant rat model of cerebral aspergillosis.** An established lethal infant rat model of cerebral aspergillosis was used (14). All animal studies strictly followed the Swiss national guidelines for the performance of animal experiments and were approved by the Animal Care and Experimentation Committee of the Canton of Bern, Switzerland (permit 7/07). Six-day-old male Wistar rats purchased from Charles River Laboratories, Sulzfeld, Germany, were housed 14 per cage with their nurse and were given food and water *ad libitum*. Rats were infected on postnatal day 11 by injection of 10 $\mu$L of the conidial suspension in saline directly into the cisterna magna, as described previously (14). The inoculum used for infection, calculated to contain 7.18 $\log_{10}$ CFU/10 $\mu$L, was verified by culture and yielded 7.17 $\pm$ 0.03 $\log_{10}$ CFU/ 10 $\mu$L ($n = 13$). Because of the relative immaturity of the immune system of infant rats, no iatrogenic immunosuppression is needed to achieve consistent infection in all animals. Intracisternal inoculation of fungal spores results in both fungal meningitis and cerebral abscesses, mimicking human CNS aspergillosis that commonly arises from hematogenous dissemination from pulmonary foci (14). After infection, weight and symptoms of CNS disease were assessed twice daily. A scoring system was used based on spontaneous activity (1 = yes, 4 = no), the ability of walking straight (1 = yes, 2 = [yes], 3 = no, 4 = not evaluable [N.E.]), snuffling (1 = yes, 2 = [yes], 3 = no), climbing out of tray on balance (1 = yes, 2 = [yes], 3 = no, 4 = N.E.), circling (1 = no, 2 = [no], 3 = yes, 4 = N.E.), dropping on one side when walking (1 = no, 3 = yes, 4 = N.E.), and rolling (1 = no, 3 = yes, 4 = N.E.). Animals with a clinical score of >16 were considered to have severe disease and were euthanized for ethical reasons by intraperitoneal (i.p.) bolus injection of 100 mg/kg pentobarbital ($n = 10$).

**Antifungal drugs and treatment.** CAS was prepared from commercial Cancidas (Merck Sharp & Dohme-Chibret AG, Horw, Switzerland) by dissolving the powder in *aqua ad iniectabilia*. Aliquots of this 5 mg/mL stock solution were stored at $-80$°C. An injectable solution was prepared by dilution of the stock solution with 0.85% saline to a final concentration of 0.1 mg/mL. Liposomal amphotericin B (L-AmB) powder (AmBisome; Gilead Sciences Sàrl, Zug, Switzerland) was reconstituted in *aqua ad iniectabili* and stored for up to 7 days at 2 to 8°C. The reconstituted solution (4 mg/mL) was filtered through a 5-$\mu$m filter according to the manufacturer's instruction and further diluted with a 5% glucose solution to a final concentration of 0.5 mg/mL. This solution was stored up to 7 days at 2 to 8°C protected from light.

Each experiment was performed with 1 or 2 litters of 14 infant rats. Animals from each litter were divided into groups of 4 to 7 animals allocated randomly to different treatment regimens. The experimenters were blinded to the treatment regimen of the animals. Treatment started 22 h after infection and was given for 10 consecutive days. Regimens were as follows: (i) CAS 1 mg/kg/d i.p. once a day (q.d.), (ii) L-AmB 5 mg/kg/d i.p. q.d., and (iii) both drugs combined at the same dose (CAS plus L-AmB) i.p. q.d. Untreated controls were given 10 mL/kg 0.85% saline i.p. q.d.

The caspofungin dose was selected in analogy with other experimental systems (11, 20, 34). In adult neutropenic rats, this dose resulted in a plasma $AUC_{0-24}$ that was lower than the $AUC_{0-24}$ found in humans treated with a daily dose of 50 mg (21). The dose of L-AmB was selected from an experimental rat model where repeated daily intravenous (i.v.) doses of 5 mg/kg L-AmB resulted in a brain tissue AmB exposure similar to the one we found (32).

**Two series of experiments were performed. (i) Series 1.** In survival studies, each treatment regimen (including saline control) was given to 21 rats from at least 3 different litters studied in separate experiments at different time points. Death or development of disease severe enough to warrant killing for ethical reasons was observed. The day animals with severe disease were killed for ethical reasons was recorded as the day of death. For animals found dead, death was assigned to the same day. The experiment was censored on day 11 postinfection when surviving rats were sacrificed by i.p. bolus injection of 100 mg/kg pentobarbital.

**(ii) Series 2.** To monitor cerebral fungal burden and to document brain drug levels over the course of the experiment, preselected animals from each of the 3 treatment regimens and saline controls were sequentially sacrificed at days 2, 3, 5, and 11 after infection (5 to 6 animals per time point from at least 3 individual experiments). Animals sacrificed for ethical reasons were excluded from this part of the study. Immediately after sacrifice, animals were perfused with 30 mL ice-cold phosphate-buffered saline (PBS). One sagittal half brain was diluted 1:1 (wt/vol) in PBS, homogenized in a tissue homogenizer, and aliquoted. Aliquots of 50 $\mu$L were serially diluted $10^{-2}$ to $10^{-4}$ in 0.85% saline and cultured on Sabouraud's agar plates with chloramphenicol and gentamicin for 24 h at 37°C before CFU were counted.

Galactomannan (GM) concentration was assessed in brain tissue by using a double-sandwich-enzyme-linked immunosorbent assay. Aliquots of homogenized brain tissue were diluted 1:40 in PBS and processed according to the manufacturer's instruction (Platelia *Aspergillus* test, Bio-Rad Laboratories AG, Reinach, Switzerland). The galactomannan index (GMI) was determined using concurrently measured internal controls and calibrators.

For later determination of drug levels, aliquots of brain homogenate were frozen at $-80$°C. Trough concentrations of CAS in brain tissue were determined 25 $\pm$ 1 h after the last drug administration. Brain homogenates were diluted 5:1 in water and analyzed by high-pressure liquid chromatography (HPLC)/ tandem mass spectrometry as described previously (35).

Trough concentrations of AmB in brain tissue were determined 25 $\pm$ 1 h after the last drug

administration using the following HPLC protocol: to 50 $\mu$L of brain homogenate, 20 $\mu$L of methanol and 125 $\mu$L of a 0.25 ng/$\mu$L solution of roxithromycin (internal standard [IS]) in methanol/acetonitrile/0.1 M zinc sulfate (8 + 1 + 1, vol/vol/vol) were added. After vortexing, the samples were centrifuged at 10,000 $\times$ $g$ at 10°C for 10 min. One hundred microliters of the supernatant was diluted with 50 $\mu$L methanol/water (1 + 1, vol/vol), and 10 $\mu$L was injected into the liquid chromatography-tandem mass spectrometry (LC-MS/MS) system (CTC PAL autosampler, Rheos 2200 HPLC pump, TSQ 7000, all Thermo Scientific, Basel, Switzerland). Chromatography was performed using a 125 by 2 mm Uptisphere $C_{18}$ 5 $\mu$m ODB column (Interchim, Montluçon, France) with 0.1% formic acid in water and acetonitrile as mobile phases mixed in a gradient mode with a flow rate of 300 $\mu$L/min. AmB and the IS were detected by selected reaction monitoring ($m/z$ 925 → 744, $m/z$ 838 → 559). A calibration curve in serum in the range of 0.025 to 1.00 mg/L was used for quantification. The precision and accuracy of the assay were monitored using a brain homogenate spiked at a concentration of 0.400 mg/L, resulting in a coefficient of variation of 7.0% and an accuracy of 97.2%.

**Statistical analysis.** Survival curves were analyzed by Kaplan-Meier analysis and compared by log rank (Mantel-Cox) test. For comparisons of continuous variables between groups, the Mann-Whitney test, Kruskal-Wallis test with Dunn's multiple-comparison test, or two-way analysis of variance with Bonferroni posttests was used. All data were expressed as mean $\pm$ standard deviation. Significance was defined as a two-tailed $P$ value of <0.05. All statistical analyses were done with GraphPad Prism software (GraphPad software Inc., San Diego, USA).

## ACKNOWLEDGMENTS

We thank Kevin Oberson and Angela Bühlmann for excellent technical assistance and Katharina Rentsch (Department of Laboratory Medicine, University of Basel, Basel, Switzerland) and Laurent Decosterd (Division of Clinical Pharmacology, University of Lausanne, Lausanne, Switzerland) for their invaluable help in measuring drug levels.

Conceived and designed the experiments: S.Z. and S.L.L. Performed the experiments: I.U. and A.A. Analyzed the data: I.U., A.A., and S.Z. Wrote the paper: I.U., A.A., and S.Z. Revised the manuscript: I.U., A.A., S.Z., and S.L.L.

This work was supported by unrestricted research grants from Merck Sharp & Dohme AG, Switzerland (https://www.msd.ch/en/home), Gilead Sciences Switzerland Sàrl (www.gilead.com/utility/global-operations/europe/switzerland), and Pfizer AG Switzerland (www.pfizer.ch). The funders had no role in study design, data collection and analysis, decision to publish, or preparation of the manuscript.

We declare that there are no competing interests.

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
