## [Reviewer comments · Microbiology Spectrum]

Microbiology Spectrum

Caspofungin cerebral penetration and therapeutic efficacy in experimental cerebral aspergillosis

Irina Ullmann, Andrea Aregger, Stephen Leib, and Stefan Zimmerli

Corresponding Author(s): Stefan Zimmerli, Bern University Hospital, Inselspital

Review Timeline:

Submission Date:	April 3, 2022
Editorial Decision:	April 5, 2022
Revision Received:	April 6, 2022
Accepted:	April 6, 2022

Editor: Dimitrios Kontoyiannis

Reviewer(s): The reviewers have opted to remain anonymous.

Transaction Report:

DOI: <https://doi.org/10.1128/spectrum.02753-21>

April 5, 2022

Dr. Stefan Zimmerli
Bern University Hospital, Inselspital
Dept. of Infectious Diseases
Freiburgstr.
Bern
Switzerland

Re: Spectrum02753-21 (Caspofungin cerebral penetration and therapeutic efficacy in experimental cerebral aspergillosis)

Dear Dr. Stefan Zimmerli:

Acknowledging more some key limitations of this otherwise sound work would strengthen this manuscript. I would suggest that the authors should modify slightly at the end (line 334) as follows (additions in CAPITAL letters):
... and IF confirmed in other animal models, ESPECIALLY IMMUNOSUPPRESSED MURINE MODELS AND BY USING ADDITIONAL ASPERGILLUS ISOLATES ACROSS A RANGE OF CAS MECs, WOULD SUGGEST that CAS merits further study..
If they make that change, I would recommend acceptance (NO need to see it again)
DPK

Link Not Available

Sincerely,

Dimitrios Kontoyiannis

Journals Department
Reviewer comments:

Staff Comments:

Preparing Revision Guidelines

Please return the manuscript within 60 days; if you cannot complete the modification within this time period, please contact me. If you do not wish to modify the manuscript and prefer to submit it to another journal, please notify me of your decision immediately so that the manuscript may be formally withdrawn from consideration by Microbiology Spectrum.

We want to thank the editor for his suggestions on how to strengthen the manuscript.

The following changes (in bold letters) were made to the manuscript Lines 338 - 341:

The efficacy of CAS in the treatment of cerebral aspergillosis documented here and **if** confirmed in other animal models, **especially immunosuppressed murine models and by using additional Aspergillus isolates across a range of CAS MECs, would suggest** that CAS merits further study as a treatment option for patients suffering from aspergillosis disseminated to the brain.

April 6, 2022

Dr. Stefan Zimmerli
Bern University Hospital, Inselspital
Dept. of Infectious Diseases
Freiburgstr.
Bern
Switzerland

Re: Spectrum02753-21R1 (Caspofungin cerebral penetration and therapeutic efficacy in experimental cerebral aspergillosis)

Dear Dr. Stefan Zimmerli:

Your manuscript has been accepted, and I am forwarding it to the ASM Journals Department for publication. You will be notified when your proofs are ready to be viewed.

Sincerely,

Dimitrios Kontoyiannis
Editor, Microbiology Spectrum
